# Selective Sweeps and Polygenic Adaptation Drive Local Adaptation along Moisture and Temperature Gradients in Natural Populations of Coast Redwood and Giant Sequoia

**DOI:** 10.3390/genes12111826

**Published:** 2021-11-19

**Authors:** Amanda R. De La Torre, Manoj K. Sekhwal, David B. Neale

**Affiliations:** 1School of Forestry, Northern Arizona University, 200 E. Pine Knoll, Flagstaff, AZ 86011, USA; manoj.kumar@nau.edu; 2Department of Plant Sciences, University of California-Davis, One Shields Avenue, Davis, CA 95616, USA

**Keywords:** selective sweeps, polygenic adaptation, GEA, climate adaptation, *Sequoiadendron giganteum*, *Sequoia sempervirens*

## Abstract

Dissecting the genomic basis of local adaptation is a major goal in evolutionary biology and conservation science. Rapid changes in the climate pose significant challenges to the survival of natural populations, and the genomic basis of long-generation plant species is still poorly understood. Here, we investigated genome-wide climate adaptation in giant sequoia and coast redwood, two iconic and ecologically important tree species. We used a combination of univariate and multivariate genotype–environment association methods and a selective sweep analysis using non-overlapping sliding windows. We identified genomic regions of potential adaptive importance, showing strong associations to moisture variables and mean annual temperature. Our results found a complex architecture of climate adaptation in the species, with genomic regions showing signatures of selective sweeps, polygenic adaptation, or a combination of both, suggesting recent or ongoing climate adaptation along moisture and temperature gradients in giant sequoia and coast redwood. The results of this study provide a first step toward identifying genomic regions of adaptive significance in the species and will provide information to guide management and conservation strategies that seek to maximize adaptive potential in the face of climate change.

## 1. Introduction

Understanding the genomic architecture of local adaptation is of great biological interest and paramount to predicting species’ responses to present and future changes in the climate. Populations’ response to shifts in trait optima because of changes in the climate will depend on the genetic basis of the trait and the population demography [1]. In traits controlled by many genes, natural selection will increase the frequency of advantageous alleles until the trait matches the new trait optimum [2]. Recent improvements in sequencing methods and genotyping approaches have allowed the detection of signatures of recent or ongoing positive selection from their molecular signature on neutral polymorphism in species genomes [2,3,4]. While most of this knowledge comes from model and domesticated species, the genomic architecture of local adaptation in natural populations of long-lived non-model species remains understudied [5].

Empirical and theoretical studies have suggested the presence of three different modes in the process of adaptation: hard sweeps, soft sweeps, and polygenic adaptation. In the classical hitchhiking model [6,7], new mutations spread rapidly to fixation due to natural selection, reducing genetic variation at neutral linked sites as they spread. These hard sweeps have received significant attention due to their distinctive signatures in the genome that are easily identified using genome-wide markers [8]. In this mutation-limited mode of adaptation, beneficial mutations are thought to be rare and therefore unlikely to be present as standing genetic variation. More recent studies have proposed that selective sweeps arising from selection on standing variation or from recurrent mutation (soft sweeps) are more common than originally thought [9,10,11]. Both hard and soft sweeps reflect only recent or on-going adaptive events since signals such as an excess of rare alleles (picked up by Tajima’s D), LD and haplotype statistics typically fade quickly (~0.01–0.1 Ne generations) [11]. In on-going adaptive events, the beneficial allele is still segregating in the population (partial sweep) until it reaches fixation and becomes a complete sweep [11]. In addition to sweeps at individual loci, relatively minor shifts in the allele frequencies of large numbers of small-effect loci (polygenic adaptation) can result in the rapid adaptation of phenotypic traits [12,13,14]. Selective sweeps and polygenic adaptation have rarely been studied together except for some theoretical [1,3,15,16,17] and a few empirical studies [18].

In contrast to genome scans that look for markers with high levels of population differentiation (Fst), multivariate genome-wide environment association (GEA) methods are more likely to identify the signals of subtle shifts in allele frequencies in weakly selected loci characteristic of polygenic adaptation [19,20,21,22]. Multivariate methods, which analyze many loci simultaneously, are better suited since they consider how sets of markers covary in response to the environment [20]. Constrained ordination procedures, such as principal component analysis (PCA) and redundancy analysis (RDA), combine the advantages of both multivariate methods and genotype-environment association procedures to detect adaptive variation based on genomic data [21,23,24]. Despite the high dimensional nature of genotype and environment, the latent factor mixed model (LFMM), which is a univariate genotype-environment association method, is also a frequently used GEA method [25].

Despite rich literature on the presence of local adaptation based on field and common garden experiments, the genomic basis of adaptation in conifer species lags behind other plant species. Non-model species attributes, such as long-generation times, large population sizes, and highly outcrossing rates, together with large genome sizes (10–30 Gb) have limited the development of genome-wide studies [26]. Genome-wide environmental association studies have been mostly based on pre-selected candidate genes in conifers [27,28,29], with just a few exceptions [19,30,31,32,33]. In all recent genome-wide studies in conifer species, adaptation to the environment occurred through many genes of small effect, characteristic of polygenic adaptation.

Giant sequoia (*Sequoiadendron giganteum*) and coast redwood (*Sequoia sempervirens*) are ancient species the fossil records of which date back to the Jurassic period, a time period in which they were more widely distributed across the northern hemisphere [34,35]. These iconic species are also some of the longest-lived trees on Earth, with records of 3200- and 2200-year-old trees, respectively [36]. Giant sequoia, the most “massive species” on Earth due to its enormous wood volume, of up to 1000 m^3^ [37], is a diploid species with a genome size of 8.125 Gbp [38]. Generally considered as an outcrossing, sexually reproducing species, giant sequoias can also reproduce vegetatively up to 20 years of age. Giant sequoias are monoecious, meaning that pollen-producing and seed-bearing cones are borne of different branches of the same individual and develop serotinous cones, which require fire to release seeds [39]. The species grows in discrete groves on the western slope of the Sierra Nevada mountains at 830–2700 m of elevation. The elevational range includes a highly disjunct range consisting of approximately 75 groves and spanning around 420 km north to south [39]. Giant sequoias depend on the melting of the accumulated snow pack from the Sierra mountains and are the most moisture demanding species of the Sierra Nevada mixed conifer forests [40].

Coast redwood (*Sequoia sempervirens*) is considered the tallest tree species on Earth, reaching up to 115 m [41]. It is a hexaploid species with a genome size of 26.5 Gbp [42]. The species is monoecious and reproduces mostly asexually but can infrequently produce flowers and seeds every 10 or 20 years. It is characterized by a particular red-colored wood and an endemic narrow natural distribution range [43]. Once extensively distributed along the Pacific Coast in Oregon and California, populations of coast redwood were severely affected by intensive logging, leading to a reduction by more than 90% of their original natural distribution by the late 19th century [44]. Both coast redwood and giant sequoia are currently listed as endangered species by the International Union for Conservation of Nature (IUCN) Red List of Threatened Species [45]. Despite their unique ecological and economic importance, the genomics of local adaptation in coast redwood and giant sequoia are insufficiently studied [38,40,46,47,48,49,50]. The objectives of this research were to dissect the genomic basis of local adaptation to climate in natural populations of giant sequoia and coast redwood by (i) locating genomic regions showing signals of natural selection that might provide information about the mode of adaptation, (ii) identifying the main environmental variables driving adaptation in the species, and (iii) understanding the main biological functions of genes associated with environmental variation.

## 2. Materials and Methods

### 2.1. DNA Extraction, Sequencing, and SNP Calling

Fresh needle tissue was collected from previously established common gardens spanning the natural distribution of the species (Figure 1). Coast redwood was collected from the hedge orchard growing in Russell Reserve (UC Field Station, Contra Costa County, CA, USA), and giant sequoia was collected from the Foresthill Divide Seed Orchard (Foresthill, CA, USA). A selected ramet was collected from each of the surviving 92 coast redwood (SESE) and 90 giant sequoia (SEGI) genets. In the lab, samples were flash-frozen in liquid nitrogen, stored in a −80 °C freezer for 48 h, and lyophilized. DNA was extracted with an E–Z 96 Plant DNA kit (Omega Biotek, Norcross, GA, USA). DNA was submitted to the UC Davis Genome Center for hybridization of baits for targeted sequencing. Exome capture was used to obtain loci from across the genome, from which SNPs were then identified and used in subsequent analyses. Exome capture reduces the complexity of large genomes by sequencing only coding regions and avoiding repeat regions of the genome. Details of the exome capture design can be found in [51]. The total capture region was 22.078 Mbp in SEGI and 37.529 Mbp in SESE. Libraries were pooled and sequenced on an Illumina NovaSeq 6000 platform. Sequencing capture raw reads were aligned against the genome assemblies of coast redwood v2.1 (treegenesdb.org/FTP/Genomes/Sese) and giant sequoia v2.0 (treegenes.db.org/FTP/Genomes/Segi) using Bowtie2 v2.2.9 [52]. Alignments were sorted and later processed in parallel using SAMtools v1.3.1 and BEDtools v2.25.0 and SNPs were called using BCFtools with default parameters [53,54]. Haplotypes were called using Genome Analysis Toolkit (GATK v.4.1.7.0) HaplotypeCaller and GenotypeGVCF [55]. Functional annotations were obtained from the giant sequoia and coast redwood reference genome annotations (treegenesdb.org) by BLASTP [56] sequence alignment against the NCBI non-redundant protein sequences database (nr) using an *e*-value < 1 × 10^−10^. BCFtools was used to merge variant call format (VCF) files of individuals for further analysis [57].

### 2.2. Population Structure

The non-model-based principal component analysis (PCA) method was used to quantify the genetic structure in SEGI and SESE. Raw genotyping data containing high levels of missing data were filtered and imputed using TASSEL v.5.2.72 [58] with the following parameters: minor allele frequency (maf) = 0.05 and maximum allele frequency (max−maf) = 0.9. A minimum count (minimum number of samples in which the site must have been scored to be included in the filtered data set) of 50 was implemented for SEGI and 30 for SESE. The imputation was applied using the LD-KNNi method [59] at TASSEL. The PCA [60] was calculated using TASSEL with a merged VCF file containing 71 individuals of SEGI and 92 individuals of SESE. The ggplot2 R package (R version 4.0.4) was used for PCA visualization. Furthermore, the Python2.x fastStructure algorithm based on a variational framework for posterior inference of K clusters was used for population structure analysis [61]. To run fastStructure, *.bed*, *.bim*, and *.fam* files were generated using PLINK v.1.9 software with filtering parameters of maf = 0.05 and max−maf = 0.4 for SEGI and SESE.

The minimum count (mac) of 50 was implemented for SEGI and 30 for SESE using PLINK [62]. The chooseK.py script at fastStructure was used in order to choose the appropriate number of model components (K) that explain the genetic structure in the dataset. Models in fastStructure were replicated 10 times, with K values from 1 to 10. The pophelper R package was performed to visualize the ancestry bar plots generated from population assignments from the fastStructure results. All R packages were carried out on R version 4.0.4 at RStudio (https://www.rstudio.com/, accessed on 15 April 2021).

### 2.3. Genome-Wide Patterns of Diversity and Differentiation

Patterns of nucleotide diversity (pi) were estimated using non-overlapping sliding windows with a step of 1000 (window-pi-step option) and a window size of 10 kbp (window-pi) in VCFtools [57]. Levels of genetic differentiation among populations were estimated using the Fixation index (Fst) method from [63], with the -weir-fst-pop option in VCFtools [57]. Data were analyzed using sliding windows with a step of 1000 (--fst-window-step option) and a window size of 10 kbp (--fst-window-size option) in VCFtools. Fst was estimated using a priori populations (groves or provenances) based on geographic location. Other population genomics statistics estimated with VCFtools included a chi-squared test of the Hardy–Weinberg equilibrium per SNP (--hardy) and an estimation of the observed and expected number of homozygous sites and inbreeding coefficient (F) per individual (--het).

### 2.4. Signatures of Positive Selection in Outlier Regions

Genomic regions of exceptionally high or low differentiation were tested for evidence of natural selection in both species using non-overlapping sliding windows of 10 kbp in VCFtools. Signatures of positive selection were considered significant in genomic regions meeting three criteria: (1) increased levels of genomic differentiation accounted by Fst values that are much larger than the average Fst across windows (after an initial analysis of overall Fst levels, an Fst > 0.1 was used as threshold), (2) reduced levels of nucleotide diversity (closer to zero), and (3) negative Tajima’s D values, indicating a skewed allele frequency spectrum toward rare alleles.

### 2.5. Linkage Disequilibrium (LD) Analysis

Linkage disequilibrium (LD) was measured based on the concept of the square of the correlation coefficient between two loci (*r*^2^), indicating the ability of the alleles present in a marker to predict the presence of alleles on a second marker located at a certain genetic distance measured in base pairs [64]. Squared correlation coefficients (*r*^2^) among all pairs of SNPs within a distance of 500 kbp were calculated using PLINK v.1.9 [62]. To quantify LD, we used TASSEL-filtered dataset and ran PLINK with the following parameters for all *r*^2^ values using --ld-window 100, --ld-window-kb 500 kb, and *r*^2^ threshold 0. Non-linear regression of pairwise *r*^2^ against the physical distance between sites (in base pairs) was performed to estimate the decay of LD in both SESE and SEGI. The decay line was calculated by nonlinear regression to the LD plot using the Hill and Weir equation [65].

### 2.6. Genome-Wide Environmental Association (GEA)

Coordinates (latitude, longitude, and elevation) representing the geographic origin of the sampled trees were collected for SESE individuals. For SEGI, only the geographic origin of the groves was known; therefore, the coordinates of the centroid of the grove polygon were employed as the geographic origin to obtain environmental data from public databases. Environmental data were obtained from WorldClim 2.0 [66] and ClimateNA [67]. All data were based on the averages for years 1962–1990 and included seasonal and annual variables. Environmental variables were selected based on previous ecological and physiological studies in the species [48,51]. Correlations among all geographic (latitude, longitude, and elevation) and environmental variables were tested in R v3.6.1. Associations among markers and variables were tested using univariate and multivariate methods. Latent factor mixed model (LFMM) version 2 [68] was used to identify univariate genome-wide environmental associations. The K values were determined using PCA, and the ridge approach was used at LFMM [68]. LFMM is computationally efficient and provides statistically optimal corrections, resulting in improved power and control for false discoveries. In the analysis, the lfmm ridge function was used in the lfmm method for estimating latent confounders (or factors). LFMM uses fitted latent factor mixed models to evaluate associations between a SNP genotype matrix and the environmental variable [69]. In addition, a redundancy analysis (RDA) multivariate GEA was implemented in the vegan R package [70]. Since RDA is a regression-based method, it can be subject to problems when using highly correlated predictors [71]; therefore, an *r* > 0.7 was used to exclude highly correlated environmental factors. The function “pairs panels” was used to visualize correlations among the environmental factors.

### 2.7. Functional Gene Annotations and Enrichment Analyses

The genomic positions of significant SNPs were used to identify the annotated genes by scanning the genomic VCF files of SEGI and SESE. Subsequently, the identified significant SNPs were annotated using annotation files downloaded from TreeGenes (https://treegenesdb.org/TripalContactProfile/588450, accessed on 1 March 2021). The annotation was confirmed using some other approaches, such as pfam [72], blastp [56] and BlastKOALA [73]. Pfam was obtained using the HMMER [74] at default parameters with *e*-value 1.0 to search for protein families. To search for similar hits, blastp ran at the expected threshold 0.05; matrix, BLOSUM 62; and database, non-redundant protein sequence (nr). BlastKOALA in KEGG [75] was performed for protein pathways and functional annotation. Identical matching genes were chosen for identifying annotation and KEGG pathways. Gene ontology (GO) enrichment analyses for biological process, molecular function, and cellular component were calculated with Blast2GO [76]. False Discovery Rate (FDR) was used for correction for multiple testing.

## 3. Results

### 3.1. Sequence Capture and SNP Datasets

A total of 630,166 and 804,682 SNPs were called for 96 SEGI and 92 SESE individuals, respectively. From them, 52,987 (9.2%) SNPs from 71 SEGI individuals and 64,358 (8.0%) SNPs from 92 SESE individuals were retained after filtering using TASSEL. The missing SNPs statistics before and after imputation for each of the SEGI and SESE individuals are reported in Appendix A, respectively. The filtered SNP dataset was retained for further analyses. The imputed dataset was only used for RDA.

### 3.2. Population Structure

To better understand the genetic structure in the datasets, a principal component (PCAs) and a fastStructure analyses were used to analyze the SNP data. The optimal number of inferred ancestral components (K) was estimated by the chooseK.py script. In giant sequoia, the output showed K = 2 as the model complexity that maximizes marginal likelihood and K = 8 as the number of model components that better explained the structure in dataset. In coast redwood, K = 1 maximized marginal likelihood and K = 3 explained the structure in dataset. Therefore, K = 2–8 was selected to describe the genetic structure of giant sequoia and K = 3 was selected for coast redwood (Figure 2). The PCA analysis showed consistent results to fastStructure, identifying eight genetic clusters in giant sequoia and three clusters in coast redwood (Appendix A). These results were consistent with previous studies in the species using more individuals but fewer molecular markers [46,48].

### 3.3. Genome-Wide Patterns of Diversity and Differentiation

Nucleotide diversity (pi) estimated using non-overlapping sliding windows averaged 0.0001842 for SESE and 0.00008 for SEGI. Average Fst was estimated as 0.01127 for SESE and 0.01593 for SEGI (Appendix A). A chi-squared test suggested that 86.5% of SNPs are in the Hardy–Weinberg equilibrium in SESE and 46.4% in SEGI. Significant levels of inbreeding (F) as estimated based on the observed and expected number of homozygous sites per individual was found in seven SEGI individuals. No evidence of inbreeding was found in SESE individuals (Appendix A).

### 3.4. Signatures of Positive Selection in Outlier Regions

Negative Tajima’s D values were found in 1390 sliding windows, each containing between 182 and 411 SNPs, with a total of 3351 SNPs distributed across the 11 chromosomes of the SEGI genome. From those, 17 genomic regions across all 11 chromosomes (with the exception of chromosomes 6 and 9), each containing one or more consecutive windows, showed elevated Fst values, higher than 0.1, and reduced nucleotide diversity, suggesting signatures of positive selection (Figure 3; Appendix A). Similarly, 336 sliding windows containing 614 SNPs across 125 scaffolds in the SESE genome also harbored negative Tajima’s D values. From these, 29 genomic regions across 21 scaffolds containing one or more sliding windows showed elevated Fst values and reduced nucleotide diversity (Appendix A).

### 3.5. Genome-Wide Linkage Disequilibrium (LD)

The pattern of LD decay was quantified based on the filtered 52,987 SNPs genotyped in 71 individuals of SEGI and 64,358 SNPs genotyped in 92 individuals of SESE. The LD decay analysis of up to 500 kb of physical distance in SEGI and SESE is shown in Appendix A. In SEGI, LD rapidly decayed between 0 and 100 kb and plateaued at ~300 kb. In contrast, LD decayed more rapidly in SESE, reaching *r^2^* levels of 0.02 at <100 k (Appendix A).

### 3.6. Genome-Wide Environmental Association (GEA)

In SEGI, associations were tested among the SNP markers and six environmental variables: mean annual precipitation (MAP), mean summer precipitation (MSP), number of frost-free days (NFFD), frost-free period (FFP), precipitation as snow (PAS), and climate moisture deficit (CMD). In SESE, tested environmental variables included annual heat moisture index (AHM), mean annual temperature (MAT), MAP, and CMD. Results indicated strong correlations among all environmental factors (Appendix A), suggesting the combined effect of groups of individual variables associated with an aspect of climate. Since RDA requires no missing data, an imputation approach was applied to the TASSEL-filtered dataset of total 52,987 SNPs for SEGI and 33,578 SNPs for SESE using the LD-KNNi method [59] at TASSEL. The multivariate RDA method identified significant associations between 292 SNP markers, matching 51 candidate genes, and six environmental variables for SEGI (Figure 4; Appendix A; Appendix A). From these, we observed 68 SNP markers associated with CMD, 11 SNPs with FFP, 83 SNPs with MAP, 61 SNPs with MSP, 44 SNPs with NFFD, and 26 SNPs with PAS in SEGI. In SESE, RDA identified 1016 significant SNPs matching 297 candidate genes. From these, 281 SNPs were associated with AHM, 217 SNPs with CMD, 218 SNPs with MAP, and 300 SNPs with MAT, as measured by their loadings along the RDA axis (Appendix A). These results identified SNPs showing associations with multivariate aspects of California’s Mediterranean climate.

LFMM results using K = 8, with an FDR threshold of 0.10, detected only three candidate SNPs in response to the environmental predictors MAP, MSP, NFFD, FFP, PAS and CMD in giant sequoia. These three candidate SNPs were at chromosome 2 and gene SEGI 30997, which were annotated as AP-2 complex subunit alpha (Appendix A). In coast redwood, a total 71 candidate SNPs (eight different genes) were identified in response to environmental predictors MAP, MAT, AHM, and CMD. Most of these candidate SNPs were located at genes SESE_067476 and SESE_121765 at scaffold 181730 and were annotated as cytochrome P450 family and respiratory burst oxidase, respectively (Appendix A). The genomic positions of significant SNPs are shown in the Manhattan plots in Appendix A. A Venn diagram analyses between GEA methods identified 10 common SNP markers in both RDA and LFMM methods in SESE and no common markers in SEGI (Appendix A; Appendix A).

Significant numbers of SNPs were involved in different metabolic and cellular processes in giant sequoia and coast redwood. For example, genes SEGI_09259 and SEGI_19226, associated with CMD environmental variable, were annotated as oxidoreductases and part of the ribosome biogenesis pathway system, respectively. Genes SEGI_01196, SEGI_17061, SEGI_02591, and SEGI_22391, associated with MAP, were involved in translation factors, MAPK signaling pathway, mRNA surveillance, and N-Glycan biosynthesis, respectively. MSP-associated genes were involved in linoleic acid metabolism, valine, leucine and isoleucine degradation, thermogenesis, and transfer RNA biogenesis. Some NFFD-associated genes were involved in N-Glycan biosynthesis and EGFR tyrosine kinase inhibitor resistance pathways (Appendix A). In SESE, major pathways of genes associated with annual heat moisture deficit were annotated as involved in glycolysis/gluconeogenesis, nitrogen metabolism, axon regeneration, glycine, serine and threonine metabolism, plant hormone signal transduction, flavonoid biosynthesis, MAPK signaling pathway, longevity regulating pathway, fatty acid degradation, and flavone and flavonol biosynthesis pathways. Genes associated with climate moisture deficit were involved in the pathways of DNA replication, Ubiquitin system, endocytosis, DNA replication, meiosis, zeatin biosynthesis, plant–pathogen interaction, circadian rhythm, glycosyltransferases, glycine, serine and threonine metabolism, MAPK signaling pathway, and monoterpenoid biosynthesis. Similarly, the major pathways of genes associated with mean annual precipitation were oxidoreductases, plant–pathogen interaction, RNA transport, amyotrophic lateral sclerosis, cyanoamino acid metabolism, propanoate metabolism, biosynthesis of secondary metabolites, calcium signaling pathway, cGMP-PKG signaling pathway, serine and threonine metabolism, pentose phosphate pathway, MMDE, C metabolism, and amino acid and nucleotide sugar metabolism (Appendix A).

### 3.7. Gene Enrichment Analyses

Results of Blast2GO suggested significant (*p*-value < 0.0001) over-representation of gene ontology in organic substance, primary, nitrogen compound, cellular, and macromolecule metabolic processes in giant sequoia (Appendix A; Appendix A). In coast redwood, over-represented GO terms included metabolic processes (cellular, primary, nitrogen compound, organic substance, phosphorous, protein, organonitrogen, cellular aromatic compound, macromolecule, and phosphate-containing compound); biosynthetic processes (organic substance, cellular, and cellular macromolecule); binding (heterocyclic compound, small molecule, carbohydrate derivative, protein, organic cyclic compound, ion, nucleic acid, nucleoside phosphate, nucleotide, ribonucleotide, purine, adenyl, phosphorous-containing groups anion, and ATP); protein modification; and catalytic, oxidoreductase and transferase activities (Appendix A; Appendix A).

## 4. Discussion

### 4.1. Selective Sweeps and Polygenic Adaptation Drive Genomic Architecture in the Species

The results of this study suggest a complex architecture of local adaptation to climate in coast redwood and giant sequoia shaped by on-going diversifying and stabilizing natural selection acting on a number of genes associated with moisture-related variables. Climate adaptation in the species is mostly driven by many genes of small effect widely distributed across the genomes of coast redwood and giant sequoia, suggesting widespread genomic changes consistent with polygenic adaptation. While recent genome-wide environmental association studies in conifer species have consistently reported polygenic adaptation [19,30,31,32,33], our study suggests a more complex genomic architecture in coast redwood and giant sequoia, with regions of the genomes undergoing polygenic adaptation, other regions with sweep-like signatures, and a smaller third group of genomic regions showing signatures of both polygenic adaptation and selective sweeps.

Despite rich literature on the population genetics and quantitative genetics models of adaptation, the question of whether selective sweeps can occur at QTLs is still poorly understood [12,13]. Chevin and Hospital [3], based on a model with one major locus and infinitely many minor loci, predicted a very low probability of selective sweeps at QTLs. In contrast, other studies have found selective sweeps using simulations of various multi-locus models [16,17] and suggested that sweeps are present even for traits under weak selection where the genomic background explains most of the variation [1]. More recently, it has been suggested that a sudden change in the environment producing a change in the phenotype optimum might produce an initial phase in which directional selection introduces small allele frequency differences that are aligned or opposed to the shift and a final longer phase maintained by stabilizing selection [77].

In our study on coast redwood, we found that 79% of the scaffolds showing signatures of selective sweeps were also associated with regions identified by multivariate RDA (Appendix A). However, only three of these scaffolds showed signatures of sweeps and polygenic adaptation in the same regions or close genomic location within the scaffolds. Additional testing suggested these results are not likely to have happened by chance (Prob = 0 with 5000 and 10,000 bootstrap). An example of these genes is the transmembrane protein SESE_103346 gene (scaffold 142490), which showed sweep signatures, was associated with mean annual temperature and annual heat moisture index, and was in relatively close proximity to other genes associated with climate moisture deficit within the same scaffold (Appendix A). The WAT1-related protein SESE_031985 gene (scaffold 195192) showed sweep signatures, was associated with mean annual precipitation and annual heat moisture index, and was in relatively close proximity to other genes associated with climate moisture deficit and annual heat moisture index within the same scaffold (Appendix A). Finally, about 10 non-coding SNPs were co-located in a genomic region (within scaffold 353338) that showed sweep signatures and was associated with mean annual temperature.

In all cases, the presence of selective sweeps, based on negative Tajima’s D; increased Fst; and reduced genetic variance, in which the advantageous alleles have not yet reach fixation (partial sweeps), suggests the presence of on-going directional or diversifying natural selection acting on genes involved in key metabolic processes, stress, transport, and reproduction in both coast redwood and giant sequoia (Appendix A). This suggests on-going or recent genome-wide changes as a consequence of rapidly changing climate conditions, which might be beneficial for the survival of the species in the long term. This adaptive response is most likely aided by standing genetic variation, as expected in long-generation species [78], despite the low levels of nucleotide diversity found in both coast redwood and giant sequoia in this study. Considering the very slow mutation rates in conifers [79], the hard-sweep-like signatures observed in this study could also have originated from standing variation since hard sweeps might sometimes originate when a single copy from standing variation is the ancestor of all beneficial alleles [11]. Further evidence on this topic will require the identification of the ancestral alleles using an outgroup species in genealogical or coalescent history studies [11].

### 4.2. Patterns of Genomic Diversity and Divergence

Highly outcrossing conifers are expected to have a rapid LD decay. It has been reported that the *r*^2^ decayed to less than 0.20 within approximately 1500 bp based on 19 candidate genes in loblolly pine [80]. In spruces, LD displayed diverse patterns among different genes or the same genes in different species, declining rapidly to half between a few base pairs and 2000 bp [81]. In Douglas-fir, LD decayed by greater than 50% over relatively short segments from *r*^2^ = 0.25 to 0.10 within 2000 bp based on 18 genes [82]. In this study, we found a rapid decay of LD in both coast redwood and giant sequoia, even though the former is mostly an asexual species. In spite of the genome-wide low levels of nucleotide diversity, coast redwood showed an elevated number of heterozygous sites per individual, based on the inbreeding coefficient F (Appendix A). This could be due to an accumulation of alleles during mitotic divisions in asexual reproduction [83]. Alternatively, a mostly asexual species, such as coast redwood, could take advantage of non-additive genetic variation in the presence of a heterozygous advantage to achieve an equilibrium where most individuals are heterozygous [84].

### 4.3. Moisture-Related Variables Drive Adaptation in the Species

Our study indicates that climate adaptation in the species is driven by many genes involved in important biological functions related to metabolic processes, stress and signaling pathways, growth, plant defense, gene expression regulation, and other mechanisms that strongly depend on variation in moisture-related environmental variables (Appendix A). In our GEA analysis, we identified 292 SNPs associated with different environmental variables, such as climate moisture deficit (CMD), mean annual precipitation (MAP), mean summer precipitation (MSP), number of frost-free days (NFFD), frost-free period (FFP), and precipitation as snow (PAS) in giant sequoia, and 1016 significant SNPs associated with annual heat moisture index (AHM), mean annual temperature (MAT), MAP, and CMD in coast redwood (Figure 4), suggesting that the study of climate adaptation requires the analysis of groups instead of individual environmental variables. Previous studies have identified modular aspects of climate adaptation in species such *Pseudotsuga menziesii* [30], *Pinus contorta* [85], and *Pinus taeda* [19]. Other GEA studies were able to identify sets of loci associated with specific climate variables, such as relative humidity and vapor water deficit, in *Abies alba* [86]; maximum temperature and annual precipitation in *Pinus cembra* and *Pinus mugo* [87]; aridity index in *Eucalyptus* [88]; mean annual temperature and precipitation in *Populus trichocarpa* [89]; and mean coldest-month temperature, extreme minimum temperature over a 30-year period, and mean annual precipitation in *P. trichocarpa* [90].

### 4.4. Functional Annotation of Genes Associated with Environmental Variables

Trees respond to changes in the environment in numerous ways, reflected as physiological, genetic, cellular, and morphological changes [91]. In this study, we identified genes associated with different environmental factors involved in many important biological processes, such as secondary metabolism (terpene, steroids, vitamins, mitochondrial, and chloroplastic), growth and reproductive development, transcription regulation, stress and signal transduction, disease resistance, and DNA processes.

Plant secondary metabolites play a crucial role in adaptation to climate change [92]. In coast redwood, we found secondary metabolism genes associated with several moisture and temperature variables. For example, SESE_068947 was associated with climate moisture deficit; SESE_021525, SESE_025033, and SESE_072616 were associated with mean annual precipitation; and SESE_038374 and SESE_075615 were associated with mean annual temperature. Other important genes involved in secondary metabolism (terpene biosynthesis) are the members of the cytochrome P450 supergene family [93]. We found cytochrome P450 genes SEGI_33526 associated with mean summer precipitation, SESE_100967 and SESE_120015 associated with annual heat moisture index, and SESE_067476 associated with climate moisture deficit and mean annual precipitation. Members of the cytochrome P450 (CYP450) superfamily have been shown to play essential roles in regulating secondary metabolite biosynthesis in *Aralia elata* (Miq.) [94]. A comprehensive genome-wide comparative expression analysis of CYP93 genes in 60 green plants revealed that CYP93 genes in dicots and monocots are preferentially expressed in the roots and tend to be induced by biotic and/or abiotic stresses [95].

Plants employ complex signaling pathways to regulate the expression of genes that allow resistance to environmental stress [96] The ubiquitin-proteasome system controls many cellular processes by degrading specific proteins and regulates key biological processes, such as hormonal signaling, growth, embryogenesis, senescence and environmental stress, and DNA repair [97,98]. Protein degradation starts when an E1 enzyme joins a Ubiquitin protein in a three-step conjugation cascade (E1 > E2 > E3) that detects specific ubiquitination signals [99]. Our study identified gene SESE_030219 annotated as E3 ubiquitin ligase, SESE_123278 as ubiquitin fusion degradation, and SESE_004990 as ubiquitin carboxyl-terminal hydrolase. E3 Ubiquitin-ligases act as central regulators of many key cellular and physiological processes, including responses to biotic and abiotic stresses in plant species [100]. Recent functional genomics studies have revealed that about 5% of the Arabidopsis genome codes for proteins are involved in the ubiquitination pathway [101].

It has been reported that calcium-dependent signaling and mitogen-activating protein kinases (MAPKs) act during abiotic stress. For instance, in *Populus euphratica*, calcium-dependent protein kinase 10 (CPK10) is expressed under drought and frost and activates both drought- and frost-responsive genes to induce stress tolerance [102]. In *Populus trichocarpa*, MAPK cascades are involved in the promotion of antioxidant stress responses and the expression of drought-related genes [103,104]. In our study, we identified genes SEGI_01196 and SEGI_29292, which were associated with mean annual precipitation and were involved in the MAPK signaling pathway. In coast redwood, MAPK signaling pathway genes included SESE_052680, associated with annual heat moisture index; SESE_027642 and SESE_105796, associated with climate moisture deficit; SESE_121765, SESE_062974, and SESE_121765, associated with mean annual precipitation; and SESE_032016, SESE_035122, and SESE_085808, associated with mean annual temperature.

Stress-inducible dehydrins belonging to the late-embryogenesis abundant (LEA) protein family are regulated by transcription factors (TFs) binding to specific responsive elements, such as ABRE, CRT/DRE/LTRE, MYB, and MYC [105]. Dehydrins are a major group of versatile proteins that play a role in many oxidative stress responses. For example, they participate in the protection of membrane integrity [106]. In our study, we identified genes SESE_085808 (transcription factor, MYC2) and SESE_066024 (MYB-like protein) (Appendix A).

The most-known R proteins in plants are the nucleotide-binding (NB) site leucine-rich repeat (LRR), which play important roles in plant defense responses to various pathogens. Plant NB-LRR proteins often have, at the N terminus, a Toll/Interleukin-1 receptor (TIR) or coiled coil (CC) domain. In plants, the functions of two additional TIR-containing protein families, TIR-NB site (TN) and TIR-unknown/random (TX), have been investigated in *Arabidopsis thaliana*, suggesting that TN proteins might act in guard complexes monitoring pathogen effectors [107,108,109]. The study of Zhang et al. [110] provided a set of 167 *NBS-LRR* genes for *Dioscorea rotundata*, which may serve as a primary resource for functional *NBS-LRR* genes against various pathogens [110]. Xu et al. [111] found that the *NBS-LRR* gene (*ZmNBS25)* in maize enhances disease resistance in rice and Arabidopsis [111]. In our study, we have identified SESE_037263, SESE_107044, and SESE_026090 as TIR/NBS/LRR disease resistance proteins that were associated with the annual heat moisture index in coast redwood.

Finally, we annotated several other genes with transporter functions, such as SESE_111042 (NRT1/PTR_family), SESE_115496 (detoxification), SESE_039889 (sugar transporter), SESE_090540 (folate-biopterin transporter), SESE_050983 and SESE_026109 (amino acid transporter), SESE_079497 (nucleobase-ascorbate transporter), and SESE_094426 (NPF family transporter) (Appendix A). The NRT1/PTR family protein NPF7.3/NRT1.5 reported in Arabidopsis is an indole-3-butyric acid transporter that is involved in root gravitropism [112]. Amino acid transporters have been identified in several model crop species, such as Arabidopsis [113], tomato [114], barley [115], maize [116] and rice [117].

## 5. Conclusions

This study provides a step toward our understanding of the genomics of adaptation to changing climates in giant sequoia and coast redwood, by identifying genomic regions and genes of adaptive significance and by reporting the geographical locations of populations or groves of importance for conservation.

## Figures and Tables

**Figure 1 genes-12-01826-f001:**
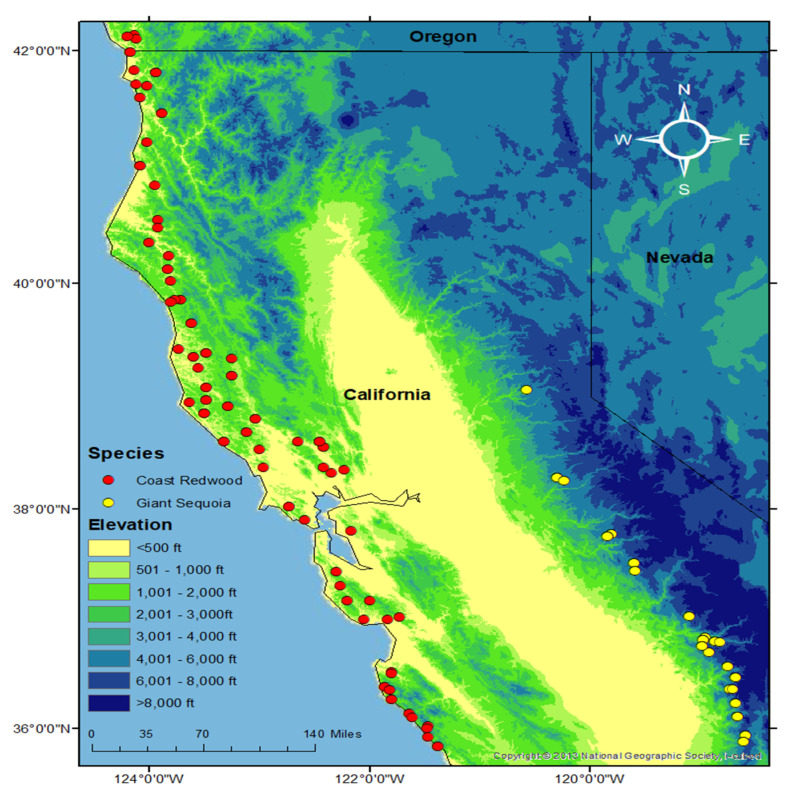
Sampling distribution of coast redwood and giant sequoia.

**Figure 2 genes-12-01826-f002:**
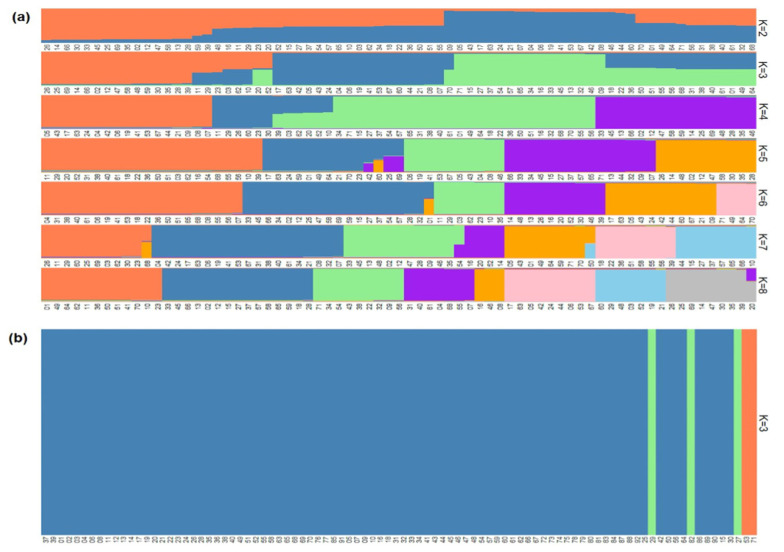
Ancestry plots of sampled individuals. Population structure plots were developed by fastStructure and R package pophelper in giant sequoia (**a**) and coast redwood (**b**).

**Figure 3 genes-12-01826-f003:**
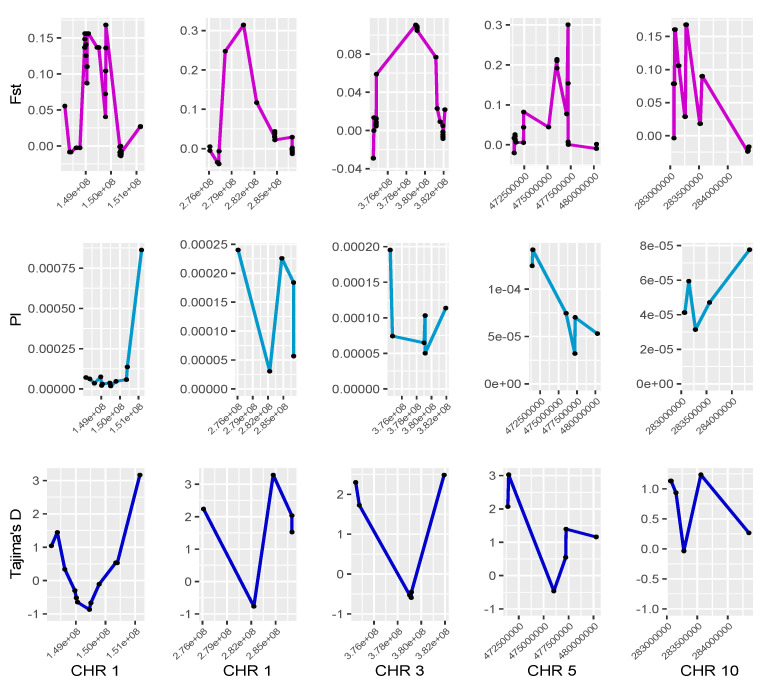
Selective sweeps in the giant sequoia genome. Patterns of nucleotide diversity (PI) and differentiation (Fst and Tajima’s D) were evaluated using sliding windows to locate genomic regions showing signatures of positive selection. Five of the longest outlier genomic regions in chromosomes 1, 3, 5 and 10 are shown.

**Figure 4 genes-12-01826-f004:**
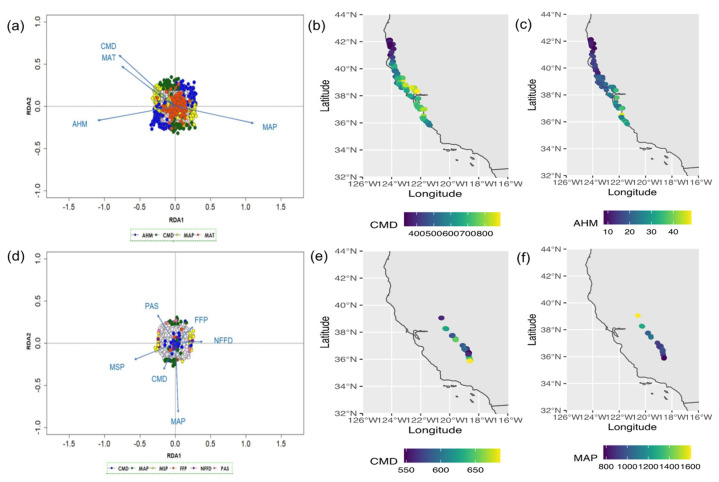
Results of multivariate RDA and distribution maps of environmental variables. Data triplots to highlight SNP loadings on RDA axes 1 and 2 (**a**) in SESE and (**d**) in SEGI. Candidate SNPs are shown as colored points with coding by most highly correlated environmental predictors. SNPs not identified as candidates (neutral SNPs) are shown in light gray. Blue vectors represent environmental predictors. Distribution maps of climate moisture deficit (CMD) and annual heat moisture index (AHM) across collected samples are displayed in (**b**) and (**c**), respectively, for SESE. CMD and mean annual precipitation (MAP) distribution maps are displayed in figures (**e**) and (**f**), respectively.

## Data Availability

Sequencing raw reads are deposited in the NCBI SRA (https://www.ncbi.nlm.nih.gov.sra, accessed on 1 March 2021) under BioSample SUB10142549.

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
