# Peer review of "Selective Sweeps and Polygenic Adaptation Drive Local Adaptation along Moisture and Temperature Gradients in Natural Populations of Coast Redwood and Giant Sequoia"

_genes, 2021, doi:10.3390/genes12111826_

Round 1

Reviewer 1 Report

Overview

De La Torre et al. here have constructed an impressive manuscript here, once again highlighting their depth of understanding of the processes involved in adaptation in forest tree species. Broadly, I think this is a fantastic piece of work that uses contemporary sequencing and analytical methods to appropriately investigate adaptation along climatic gradients in two iconic forest tree species. The authors present strong evidence for the presence of genomic regions showing signatures of positive selection (selective sweeps), as well as good evidence for the presence of multilocus molecular signatures representing polygenic adaptation. I think the overall framing and style of the manuscript is of high quality. While I have a few broader comments and a number of minor suggestions to improve the manuscript, I cannot see any major revisions being required.

Broad comments

I think the aims and hypotheses could be expanded/improved upon in the final paragraph of the Introduction. What do the authors expect they will discover and why? For example, do they expect to find hundreds of genes contributing to adaptation to a single climatic variable? Do they have some idea of the power of their sequencing methods to identify polygenic adaptation when dozens or hundreds of loci may be involved? Are the biological functions of all genes able to be identified using the reference databases?

I think Section 2.1 (DNA extraction, sequencing and SNP calling) needs a little more information about the sampling design and sequencing methods used. The paper that the authors cite (Line 123) regarding details of the exon capture process does not appear to be published yet (states “under review” in the reference list), so I am unable to check the details of the sequencing protocol. WRT sampling design, how many localities were sampled for each species? The manuscript mentions 75 groves of giant sequoia, with 90 individuals sequenced, so presumably there’s some groves represented by more than 1 individual? A supplementary table showing numbers for each provenance name would be helpful in this regard. I’m also guessing that exome capture was chosen because sequencing just the exome reduces the complexity of the large genomes of the conifers being analysed while also avoiding repeat regions? If so, please state this in the manuscript. Could the authors please also add a sentence (e.g. around Line 120) into this paragraph simply stating broadly that exome capture was used to obtain loci from across the genome from which SNPs were then identified and used in the subsequent analyses? Another broad sentence somewhere in the Methods (or Results) that describes the basics of what was actually achieved would be helpful; e.g. ‘we sequenced all exons from both species, resulting in sequence data of XX nuclear loci with an average length of ZZ bp’. Other information needed: (1) were unmapped reads removed? (2) what were the quality score thresholds used for base calls and mapping quality? Perhaps some of this information could be added to the Supplementary material if the authors want to save space in the main document.

Choosing K values is always a tricky process. I like that the authors have suggested that a range of K values are appropriate for understanding population genomic structure in giant sequoia, indicating the inherent hierarchical nature of population structure. However, it seems like the most fine-scale population structure (K=8) was selected for the giant sequoia LFMM analysis, which is probably leading to the small number of genotype-environment associations that were found for this species via LFMM (3 SNPs, compared to 71 SNPs for coast redwood, which used K=2). I suspect it would be worth re-running the LFMM for this species at a lower K value, and possibly including both sets of results to compare the difference, unless the authors can strengthen the justification for the use of K=8. Getting this K selection right is an important component of the results for this manuscript, as the low number of associations from LFMM supports the notion that polygenic adaptation is the more common mode of adaptation.

Minor comments/suggestions

Line 53: Suggest rewording this sentence; “rapid phenotypic variation” doesn’t sound right to me. Perhaps something like “In addition to sweeps at individual loci, relatively minor shifts in the allele frequencies of large numbers of small-effect loci (‘polygenic adaptation’) can result in rapid adaptation of polygenic traits.”

Line 78: change “Jurassic time, period” to “Jurassic, a time period”

Line 133: Perhaps better to use the full term for VCF the first time you mention it; i.e. change “merge vcfs files” to “merge variant call format (VCF) files”. Also capitalise the acronym throughout e.g. Line 142?

Line 134: correct spelling of “Population Sructure” to “Population Structure”

Line 142: optional change “vcf” to “VCF”.

Line 143: fix wording of sentence regarding visualisation (“The R package ggplot2 were carried out in R version”)

Line 161: While the actual values of Fst look reasonable for forest tree species, do the authors believe that the differentiation values are accurate when sample sizes per population are (presumably) small? E.g. if there’s only 1-3 individuals per ‘population’, isn’t Fst going to be poorly estimated? I’m guessing here that Fst was calculated using a-priori populations (e.g. groves) not genomically identified populations (identified in fastStructure), but please state in the manuscript which of these was used.

Line 208: The methods state than an r2>0.7 was used to exclude highly correlated environmental factors for the RDA analysis, yet the results mention strong correlations between variables (Line 285) and it appears as though highly correlated (r>0.9) variables were used in the RDA (Figures 4 and S4)?

Line 223: Use the full name for FDR (used only twice in manuscript).

Line 257: Why in this paragraph do the authors only report the findings from one of the two species?

Line 280: Could the authors include a sentence somewhere in this paragraph stating why they chose the particular environmental variables that were used? Presumably these variables have previously been shown to be important?

Line 285: include “(r > 0.5)” or something after the word “strong” to state level of correlation that is considered strong. Also check wording of highly correlated versus strongly correlated throughout the manuscript and keep consistent.

Line 287-289: I can’t find any missing data imputation method that’s done via the vegan package. Could the authors provide more detail in the manuscript (also preferably move to Methods section) about how they imputed missing data, e.g. presumably the mode was used to complete missing SNPs with the most common genotype from across all samples? Not the most common allele from the ancestral population or something? Also, will imputing reasonably large amounts of missing data (as appears to have been done for these datasets) result in a reduction in the number of significant SNP-climate associations (e.g. because a common allele is used to replace a potentially rarer allele at all geographic locations)?

Line 298: use the full name for GIF (it appears to only be used once in the manuscript).

Line 289-306: This article (https://onlinelibrary.wiley.com/doi/full/10.1111/mec.14549) may help with reporting of results for GEAs; specifically note that: “authors should explicitly report the total number of significant associations, the number of unique SNPs that have associations, the total number of associations per SNP, as well as which SNPs have significant associations with which environmental factor”.

Lines 300 and 302: in the previous paragraph the authors use ‘SESE’ and ‘SEGI’ whereas here they use ‘giant sequoia’ and ‘coast redwood’. Worth checking consistency of usage throughout manuscript.

Line 406-408: Suggest rewording this sentence slightly; perhaps just needs the ‘and’ removed or a ‘were’ inserted before the ‘co-located’.

Line 414: change ‘suggest’ to ‘suggests’, and change ‘wide genomic’ to ‘genome-wide’.

Line 414: Can the authors discuss, in or near this paragraph, what they think the effect of population bottlenecks would be on Tajima’s D in relation to Fst and pi? Would older bottlenecks not also act to reduce Tajima’s D below zero, increase Fst and reduce nucleotide diversity, similarly to a selective sweep? I’m guessing that a ‘recent’ bottleneck, e.g. caused by logging over the past couple of hundred years, would not be observed in the Tajima’s D for samples in this study?

Line 433: Could this excess of heterozygous loci be the result of selection against homozygote individuals within a stand (heterozygote advantage)? Presumably selection would just prune out homozygotes as a cohort grows over hundreds of years. There may also be strong differences in heterozygosity/FIS in genic and non-genic parts of the genome (https://onlinelibrary.wiley.com/doi/full/10.1111/mec.15894; figure 2).

Figures and tables:

Supplementary Figure S1: Please improve wording of second sentence of caption: “PCA was carried out at TASSEL and visualization in R for genotype”

Figure 2: How are individuals ordered in these fastStructure plots? Could you state in the legend whether they are ordered in a north-south direction, etc?

Figure 3 (caption): States that four genomic regions are shown, but there are five regions displayed –one of the Chr1 regions could be removed from the figure and the plots for the other regions widened to simplify the figure and improve ease of interpretation.

Figure 4 (caption Line 313): should this caption read “(a) in SESE and (d) in SEGI” ?

Author Response

Response to reviewers- Genes-1412322

Reviewer 1

Overview

De La Torre et al. here have constructed an impressive manuscript here, once again highlighting their depth of understanding of the processes involved in adaptation in forest tree species. Broadly, I think this is a fantastic piece of work that uses contemporary sequencing and analytical methods to appropriately investigate adaptation along climatic gradients in two iconic forest tree species. The authors present strong evidence for the presence of genomic regions showing signatures of positive selection (selective sweeps), as well as good evidence for the presence of multilocus molecular signatures representing polygenic adaptation. I think the overall framing and style of the manuscript is of high quality. While I have a few broader comments and a number of minor suggestions to improve the manuscript, I cannot see any major revisions being required.

>> We thank the reviewer for the positive comments and interesting suggestions. We have made every effort to address them.

Broad comments

I think the aims and hypotheses could be expanded/improved upon in the final paragraph of the Introduction. What do the authors expect they will discover and why? For example, do they expect to find hundreds of genes contributing to adaptation to a single climatic variable? Do they have some idea of the power of their sequencing methods to identify polygenic adaptation when dozens or hundreds of loci may be involved? Are the biological functions of all genes able to be identified using the reference databases?

>> We thank the reviewer for these questions. Although they are very interesting, we believe discussing about these topics might take a significant space in the manuscript and might be out of the scope of this study. For example, regarding the last question, we can say that functional annotation of conifer genomes has gotten much better in recent years, but it is still not at the level of model plant species. However, just ten years ago, all genes’ annotations were compared to Arabidopsis, knowing that 300 million years of evolution between angiosperms and gymnosperms have surely caused some significant changes in the genes, gene families and genomes. So, perhaps the answer to the reviewer question is that, yes, we were able to identify most gene functional annotations, but some of them matched to “uncharacterized proteins” and some did not have any match.  It is important to mention that while we advance on our understanding of the coding regions of the genomes, most conifers genomes have between 70-80% repetitive regions, from which we know very little about.

I think Section 2.1 (DNA extraction, sequencing and SNP calling) needs a little more information about the sampling design and sequencing methods used. The paper that the authors cite (Line 123) regarding details of the exon capture process does not appear to be published yet (states “under review” in the reference list), so I am unable to check the details of the sequencing protocol.

>> A biorxiv pre-print containing details on the sequencing methods has been made available and it is now cited in this manuscript, please see below for the full citation:

“De La Torre AR, Sekhwal MK, Puiu D, Salzberg SL, Scott AD, Allen B, Neale DB, Chin ARO, Buckley TN. 2021. Genome-wide association identifies candidate genes for drought tolerance in coast redwood and giant sequoia. BioRxiv,doi: https://doi.org/10.1101/2021.10.25.465813

WRT sampling design, how many localities were sampled for each species? The manuscript mentions 75 groves of giant sequoia, with 90 individuals sequenced, so presumably there’s some groves represented by more than 1 individual? A supplementary table showing numbers for each provenance name would be helpful in this regard.

>> The giant sequoia sample collection included 1-7 individuals from 24 different groves. The name of these groves and the PCA results can be observed in Figure S1.

I’m also guessing that exome capture was chosen because sequencing just the exome reduces the complexity of the large genomes of the conifers being analysed while also avoiding repeat regions? If so, please state this in the manuscript. Could the authors please also add a sentence (e.g. around Line 120) into this paragraph simply stating broadly that exome capture was used to obtain loci from across the genome from which SNPs were then identified and used in the subsequent analyses?

>>The following paragraph has been added to the Materials and Methods section of the manuscript:

“Exome capture was used to obtain loci from across the genome, from which SNPs were then identified and used in subsequent analyses. Exome captures reduces the complexity of large genomes by only sequencing coding regions and avoiding repeat regions of the genome.”

Another broad sentence somewhere in the Methods (or Results) that describes the basics of what was actually achieved would be helpful; e.g. ‘we sequenced all exons from both species, resulting in sequence data of XX nuclear loci with an average length of ZZ bp’. Other information needed: (1) were unmapped reads removed? (2) what were the quality score thresholds used for base calls and mapping quality? Perhaps some of this information could be added to the Supplementary material if the authors want to save space in the main document.

>> A bioRxiv pre-print containing details on the sequencing methods and results has been made available and it is now cited in this manuscript, please see below for the full citation:

“De La Torre AR, Sekhwal MK, Puiu D, Salzberg SL, Scott AD, Allen B, Neale DB, Chin ARO, Buckley TN. 2021. Genome-wide association identifies candidate genes for drought tolerance in coast redwood and giant sequoia. BioRxiv,doi: https://doi.org/10.1101/2021.10.25.465813”

Choosing K values is always a tricky process. I like that the authors have suggested that a range of K values are appropriate for understanding population genomic structure in giant sequoia, indicating the inherent hierarchical nature of population structure. However, it seems like the most fine-scale population structure (K=8) was selected for the giant sequoia LFMM analysis, which is probably leading to the small number of genotype-environment associations that were found for this species via LFMM (3 SNPs, compared to 71 SNPs for coast redwood, which used K=2). I suspect it would be worth re-running the LFMM for this species at a lower K value, and possibly including both sets of results to compare the difference, unless the authors can strengthen the justification for the use of K=8. Getting this K selection right is an important component of the results for this manuscript, as the low number of associations from LFMM supports the notion that polygenic adaptation is the more common mode of adaptation.

>> We believe our population structure estimates are accurate for the species. We compared our population structure analyses with previous studies in the species and found very similar results. For example, Breidenbach et al. 2020, found that K=2 for coast redwood populations using a small number of SSR markers; and DeSilva & Dodd (2020) found a K=9 using 1364 SNPs in giant sequoia. The sampling they used in that study was different than ours, therefore, the small difference in K could be attributed to that. These citations have been added to the “Population Structure” subheading of the Results section of the manuscript. This now reads: “These results were consistent with previous studies in the species using more individuals but fewer molecular markers [45, 47].” For these reasons, we don’t believe than lowering the K number for LFMM analyses is a good idea for giant sequoia, as unaccounted population structure might lead to an increase in false positives in the GEA results.

Minor comments/suggestions

Line 53: Suggest rewording this sentence; “rapid phenotypic variation” doesn’t sound right to me. Perhaps something like “In addition to sweeps at individual loci, relatively minor shifts in the allele frequencies of large numbers of small-effect loci (‘polygenic adaptation’) can result in rapid adaptation of polygenic traits.”

>>This sentence has been modified as suggested by the reviewer,

Line 78: change “Jurassic time, period” to “Jurassic, a time period”

>>This sentence has been changed.

Line 133: Perhaps better to use the full term for VCF the first time you mention it; i.e. change “merge vcfs files” to “merge variant call format (VCF) files”. Also capitalise the acronym throughout e.g. Line 142?

>> This sentence has been changed.

Line 134: correct spelling of “Population Sructure” to “Population Structure”

>>This subtitle has been corrected.

Line 142: optional change “vcf” to “VCF”.

>>All vcfs have been capitalized throughout the manuscript.

Line 143: fix wording of sentence regarding visualisation (“The R package ggplot2 were carried out in R version”)

>>This sentence has been rephrased and now reads: “The ggplot2 R package (R version 4.0.4) was used for PCA visualization.”

Line 161: While the actual values of Fst look reasonable for forest tree species, do the authors believe that the differentiation values are accurate when sample sizes per population are (presumably) small? E.g. if there’s only 1-3 individuals per ‘population’, isn’t Fst going to be poorly estimated? I’m guessing here that Fst was calculated using a-priori populations (e.g. groves) not genomically identified populations (identified in fastStructure), but please state in the manuscript which of these was used.

>> A new sentence has been added to the “Genome-wide patterns of diversity and differentiation in the Methods section of the manuscript:

“Fst was estimated using a priori populations (groves or provenances) based on geographic location.”

Line 208: The methods state than an r2>0.7 was used to exclude highly correlated environmental factors for the RDA analysis, yet the results mention strong correlations between variables (Line 285) and it appears as though highly correlated (r>0.9) variables were used in the RDA (Figures 4 and S4)?

>> Lines 285-286 in the original manuscript referred to correlations among all variables (Figure 4 is also reflecting that). Only a few of those, that were not highly correlated were selected for RDA analyses.

Line 223: Use the full name for FDR (used only twice in manuscript).

>>The full name of FDR has been included in the manuscript.

Line 257: Why in this paragraph do the authors only report the findings from one of the two species?

>> The section “signatures of positive selection in outlier regions” (lines 255-264 of the original manuscript) reports the results of both coast redwood and giant sequoia.

Line 280: Could the authors include a sentence somewhere in this paragraph stating why they chose the particular environmental variables that were used? Presumably these variables have previously been shown to be important?

>> The Genome-wide environmental association (GEA) section of the Materials and Methods has been modified and now reads:

“Coordinates (latitude, longitude and elevation) representing the geographic origin of the sampled trees were collected for SESE individuals. For SEGI, only the geographic origin of the groves was known; therefore, the coordinates of the centroid of the grove polygon was employed as geographic origin to obtain environmental data from public databases. Environmental data was obtained from WorldClim2.0 [65] and ClimateNA [66]. All data was based on the averages for years 1962-1990, and included seasonal and annual variables. Environmental variables were selected based on previous ecological and physiological studies in the species [47, 50].”

Line 285: include “(r > 0.5)” or something after the word “strong” to state level of correlation that is considered strong. Also check wording of highly correlated versus strongly correlated throughout the manuscript and keep consistent.

>> This sentence has been modified and now reads “These results identified SNPs showing associations with multivariate aspects of California’s Mediterranean climate.”

Line 287-289: I can’t find any missing data imputation method that’s done via the vegan package. Could the authors provide more detail in the manuscript (also preferably move to Methods section) about how they imputed missing data, e.g. presumably the mode was used to complete missing SNPs with the most common genotype from across all samples? Not the most common allele from the ancestral population or something? Also, will imputing reasonably large amounts of missing data (as appears to have been done for these datasets) result in a reduction in the number of significant SNP-climate associations (e.g. because a common allele is used to replace a potentially rarer allele at all geographic locations)?

>> This sentence has been modified and now reads: “Since RDA requires no missing data, an imputation approach was applied to the TASSEL-filtered dataset of total 52,987 SNPs for SEGI and 33,578 SNPs for SESE using LD-KNNi method [58] at TASSEL”. More information about imputation can be found in the Methods section”.

Line 298: use the full name for GIF (it appears to only be used once in the manuscript).

>> This term has been removed from the manuscript.

Line 289-306: This article (https://onlinelibrary.wiley.com/doi/full/10.1111/mec.14549) may help with reporting of results for GEAs; specifically note that: “authors should explicitly report the total number of significant associations, the number of unique SNPs that have associations, the total number of associations per SNP, as well as which SNPs have significant associations with which environmental factor”.

>> This paragraph was edited and now reads:

“The multivariate RDA method identified significant associations between 292 SNP markers, matching 51 candidate genes, and 6 environmental variables for SEGI (Figure 4, Supplementary Table S5 and Figure S5). From these, we observed 68 SNPs markers associated with CMD, 11 SNPs with FFP, 83 SNPs with MAP, 61 with MSP, 44 SNPs with NFFD and 26 SNPs with PAS in SEGI. In SESE, RDA identified 1016 significant SNPs matching 297 candidate genes. From these, 281 SNPs were associated with AHM, 217 SNPs with CMD, 218 SNPs with MAP and 300 SNPs with MAT as measured by their loadings along the RDA axis (Supplementary table S6).”

Lines 300 and 302: in the previous paragraph the authors use ‘SESE’ and ‘SEGI’ whereas here they use ‘giant sequoia’ and ‘coast redwood’. Worth checking consistency of usage throughout manuscript.

>>This has been modified and species names has been replaced by acronyms.

Line 406-408: Suggest rewording this sentence slightly; perhaps just needs the ‘and’ removed or a ‘were’ inserted before the ‘co-located’.

>> This sentence has been modified and now reads: “Finally, about ten non-coding SNPs were co-located in a genomic region (within scaffold 353338) that showed sweep signatures and was associated with mean annual temperature.”

Line 414: change ‘suggest’ to ‘suggests’, and change ‘wide genomic’ to ‘genome-wide’.

>> This sentence has been modified.

Line 414: Can the authors discuss, in or near this paragraph, what they think the effect of population bottlenecks would be on Tajima’s D in relation to Fst and pi? Would older bottlenecks not also act to reduce Tajima’s D below zero, increase Fst and reduce nucleotide diversity, similarly to a selective sweep? I’m guessing that a ‘recent’ bottleneck, e.g. caused by logging over the past couple of hundred years, would not be observed in the Tajima’s D for samples in this study?

>> We thank the reviewer for this is an excellent question. Dodd & Desilva (2016) used SSR markers and Bayesian computation and inferred a demographic decline that began about 2.3 Mya for giant sequoia, probably coincident with major climatic changes at the end of the tertiary. The same study suggests that changes in distribution and range shifts occurred slowly and therefore would not influence the genetic signature of demographic change. Both coast redwood and giant sequoia were subject to strong bottleneck in the late 19th century, this seems like a long time ago but for these species is just a short amount of time in their very long lifespans. With that said, we don’t believe Tajima’ D would pick up some signals from these recent bottlenecks; and we are not aware of older bottlenecks in any of the species.

Line 433: Could this excess of heterozygous loci be the result of selection against homozygote individuals within a stand (heterozygote advantage)? Presumably selection would just prune out homozygotes as a cohort grows over hundreds of years. There may also be strong differences in heterozygosity/FIS in genic and non-genic parts of the genome (https://onlinelibrary.wiley.com/doi/full/10.1111/mec.15894; figure 2).

>>We thank the reviewer for this interesting comment. We have added the following sentence to our Discussion section:

“In spite of the genome-wide low levels of nucleotide diversity, coast redwood showed elevated number of heterozygous sites per individual, based on the inbreeding coefficient F (Supplementary figure S2). This could be due to an accumulation of alleles during mitotic divisions in asexual reproduction [81]. Alternatively, a mostly asexual species such as coast redwood could take advantage of non-additive genetic variation in the presence of heterozygous advantage to achieve an equilibrium where most individuals are heterozygous [82].”

And we added the reference: Peck & Waxman. 2000. What’s wrong with a little sex?. J. Evol. Biol.  

 Figures and tables:

Supplementary Figure S1: Please improve wording of second sentence of caption: “PCA was carried out at TASSEL and visualization in R for genotype”

>> This figure legend has been modified and now reads: “Population structure in SEGI (a) and SESE (b) based on principal component analysis (PCA) results. PCA was carried out at TASSEL and plots were performed in R.”

Figure 2: How are individuals ordered in these fastStructure plots? Could you state in the legend whether they are ordered in a north-south direction, etc?

>> Individuals are ordered by ancestry in the FastStructure plots.

Figure 3 (caption): States that four genomic regions are shown, but there are five regions displayed –one of the Chr1 regions could be removed from the figure and the plots for the other regions widened to simplify the figure and improve ease of interpretation.

>>This figure legend has been modified.

Figure 4 (caption Line 313): should this caption read “(a) in SESE and (d) in SEGI” ?

>>We thank the reviewer for this observation. This figure legend has been corrected.

Reviewer 2 Report

This study used genotype–environment association methods, and a selective sweep analysis to identify genomic regions of potential adaptive importance, and showed associations of SNPs to moisture variables and mean annual temperature.

A total of 92 trees for coast redwood (SESE) and 90 trees of giant sequoia (SEGI) were used to study population structure, genome-wide patterns of diversity and differentiation, and genome-wide environmental association. I have two main concern: (1) population size may be too small for such association study, and (2) the nucleotide diversity estimated at 0.0001842 for SESE and 0.00008 for SEGI was much (about 100 times) lower than many published estimates for other conifers that is between 0.005- 0.015. Can we explain why these two species had such low nucleotide diversity? Very low diversity would also affect LD, local adaption and association between genotypes and environment and between genotypes and phenotypes. These need to be discussed.

These two species are probably mal-adapted to current climates of California with such narrow distributions and low level of sexual regeneration, and are listed as endangered species. Perhaps, study of slow adaptation of these two species to climate changes due to low genetic diversity would be more interesting.

The genome-wide environmental association was presented, what about genome-wide phenotype association for traits that are sensitive to moisture (drought resistance) and to temperature (cold resistance).

The gene functional annotation occupied a large proportion of the manuscript including the discussion. This may be reduced or re-focused.

Other comments

L. 36. “the genomic architecture of local adaptation in natural populations of long-lived non-model species remains largely unknown”. This is not true, for example, there are a dozen studies in Norway spruce and other spruce, I only listed a few publications here. Heuertz et al. 2006, Genetics 174:2095-2105; Chen et al, 2012 Genetics 191:865-881; Chen et al Genetics 2014, 197:1025-1038; Chen et al. G3 2016 6:1979-89; Wang et al. 2020 GBE 12: 3803–3817.

L. 239, 90 trees can be attributed into from 2-8K population structure. Is there enough population size for each cluster to have a meaningful population structure?

L. 267. What do you mean - the genetic forces that structure genome?

Table 1, occupying more than 2 pages in the manuscript should be moved to supplementary.

Table 2, occupying 9 pages in the manuscript should be moved to supplementary.

L. 374. it stated that diversifying and stabilizing natural selection acting on a number of genes associated with moisture-related variables. What is the evidence? Were genes that are associated with moisture tested for selection?

L. 380. I cannot see a more complex genomic architecture in coast redwood and giant sequoia than other tree species with such low genetic diversity in both species. Have you compared with other publications such as Norway spruce, white/black spruce which had more comprehensive studies on complex genomic architecture?

L. 411-418, giving only three of these scaffolds showed signatures of sweeps and polygenic adaptation in the same regions or very close genomic location within the scaffolds, the suggestion that “the presence of on-going directional or diversifying natural selection acting on genes involved in key metabolic processes, stress, transport and reproduction in both coast redwood and giant sequoia ….” may be too stretched.

Author Response

Response to reviewers- Genes-141232

Reviewer 2

This study used genotype–environment association methods, and a selective sweep analysis to identify genomic regions of potential adaptive importance, and showed associations of SNPs to moisture variables and mean annual temperature.

A total of 92 trees for coast redwood (SESE) and 90 trees of giant sequoia (SEGI) were used to study population structure, genome-wide patterns of diversity and differentiation, and genome-wide environmental association. I have two main concern: (1) population size may be too small for such association study, and (2) the nucleotide diversity estimated at 0.0001842 for SESE and 0.00008 for SEGI was much (about 100 times) lower than many published estimates for other conifers that is between 0.005- 0.015. Can we explain why these two species had such low nucleotide diversity? Very low diversity would also affect LD, local adaption and association between genotypes and environment and between genotypes and phenotypes. These need to be discussed.

>>Regarding small sample sizes:

We thank the reviewer for this comment. We are aware of the limitations of using small datasets in a genome-wide association analysis, therefore we use several methods and models in our GEA analyses (LFMM and RDA). In addition, we have measured 10 drought related traits and perform univariate and multivariate GWAS analyses in TASSEL and GEMMA. Common associated genes were found between the two manuscripts (GWAS vs. GEA). The GWAS paper is still under review in another journal, but a pre-print has been made available to reviewers (please see citation below).

The use of several methods and software with the same datasets is a way of cross-validation and a more accurate detection of false positives and has been widely used in previous GEA/GWAS studies in tree species. The identification of candidate genes is extremely important in conifer species where the genomic basis of phenotypic variation is broadly understudied. In the case of giant sequoia and coast redwood, there is no previous information about the presence of any candidate gene in relation to any phenotypic trait or environmental variable. We hope that our study, together with the sequencing of the reference genomes, will pave the way for additional studies that will fill this gap in our understanding of phenotypic variation and local adaptation in long-generation North American conifer species.

>>Regarding low levels of nucleotide diversity:

Low levels of nucleotide diversity can be attributed to several causes including the type of reproduction in which asexual reproduction is predominant in coast redwood and significant in giant sequoia; the presence of clones in the species; low gene flow among groves; and others. Some of these explanations are discussed in the SESE genome paper (Neale et al. 2021); as well as some comparisons with other asexual plants. Dodd & DeSilva (2016) using Bayesian computation, attributed low genetic diversity in giant sequoia to a demographic decline that began 2.3 Mya, coincident with major climatic changes at the end of the tertiary.

These two species are probably mal-adapted to current climates of California with such narrow distributions and low level of sexual regeneration, and are listed as endangered species. Perhaps, study of slow adaptation of these two species to climate changes due to low genetic diversity would be more interesting.

>>We don’t have any evidence of maladaptation or slow adaptation in any of the species in this study, therefore we are unable to comment in that regard.

The genome-wide environmental association was presented, what about genome-wide phenotype association for traits that are sensitive to moisture (drought resistance) and to temperature (cold resistance).

>> A GWAS study of 10 drought-related traits and the SNPs obtained in this study is under review in another scientific journal. A biorxiv pre-print has been made available and it is now cited in this manuscript, please see below for the full citation:

“De La Torre AR, Sekhwal MK, Puiu D, Salzberg SL, Scott AD, Allen B, Neale DB, Chin ARO, Buckley TN. 2021. Genome-wide association identifies candidate genes for drought tolerance in coast redwood and giant sequoia. BioRxiv,doi: https://doi.org/10.1101/2021.10.25.465813”

The gene functional annotation occupied a large proportion of the manuscript including the discussion. This may be reduced or re-focused.

>> We appreciate the comment, but we would like to keep this important part of the Discussion in the main text due to its importance to the paper. The functional annotation of candidate genes is a very important aspect of this paper that is only possible because of the recent sequencing of the genomes of coast redwood and giant sequoia. This sets our paper apart from earlier genetic/ genomic studies in the species, in which genes were hardly identified and couldn’t be annotated.

Other comments

  1. 36. “the genomic architecture of local adaptation in natural populations of long-lived non-model species remains largely unknown”. This is not true, for example, there are a dozen studies in Norway spruce and other spruce, I only listed a few publications here. Heuertz et al. 2006, Genetics 174:2095-2105; Chen et al, 2012 Genetics 191:865-881; Chen et al Genetics 2014, 197:1025-1038; Chen et al. G3 2016 6:1979-89; Wang et al. 2020 GBE 12: 3803–3817.

>> This sentence has been modified and now reads: “While most of this knowledge comes from model and domesticated species, the genomic architecture of local adaptation in natural populations of long-lived non-model species remains understudied”.

>> The reasoning behind this sentence is to compare model and non-model tree species, and not to diminish or ignore previous contributions to the field of forest genomics. We hope that the modified sentence makes this clear to the journal’s audience.

  1. 239, 90 trees can be attributed into from 2-8K population structure. Is there enough population size for each cluster to have a meaningful population structure?

>> We believe our population structure estimates are accurate for the species. We compared our population structure analyses with previous studies in the species and found very similar results. For example, Breidenbach et al. 2020, found that K=2 for coast redwood populations using a small number of SSR markers; and DeSilva & Dodd (2020) found a K=9 using 1364 SNPs in giant sequoia. The sampling they used in that study was different than ours, therefore, the small difference in K could be attributed to that. These citations have been added to the “population Structure” subheading of the Results section of the manuscript. This now reads: “These results were consistent with previous studies in the species using more individuals but fewer molecular markers [45, 47].”

  1. 267. What do you mean - the genetic forces that structure genome?

>> This sentence has been removed.

Table 1, occupying more than 2 pages in the manuscript should be moved to supplementary.

>> We appreciate the comment, but we would like to keep Table 1 in the main text due to its importance to the paper.

Table 2, occupying 9 pages in the manuscript should be moved to supplementary.

>> We appreciate the comment, but we would like to keep Table 1 in the main text due to its importance to the paper. We have deleted all genes annotated as “uncharacterized proteins” to reduce the length of the table.

  1. 374. it stated that diversifying and stabilizing natural selection acting on a number of genes associated with moisture-related variables. What is the evidence? Were genes that are associated with moisture tested for selection?

>> Yes, they were. This paragraph summarizes the results of the combined GEA analyses and selective sweeps, in which several genes associated with moisture were also identified in the sweep analyses. More information can be found in lines 395-397 of the original manuscript that states that in coast redwood, 79% of the scaffolds showing signatures of selective sweeps, were also associated with regions identified by the RDA analyses.

  1. 380. I cannot see a more complex genomic architecture in coast redwood and giant sequoia than other tree species with such low genetic diversity in both species. Have you compared with other publications such as Norway spruce, white/black spruce which had more comprehensive studies on complex genomic architecture?

>>Again, the comparison here is between conifers and non-conifers, and it is meant for an audience that is not within the field of forest genomics. The journal Genes has a broad audience and therefore papers should be aimed at them. We believe that many conifers have complex genomic architecture, we have published about this in several other papers in species such as sugar pine, loblolly pine, and Douglas fir. We believe that is also the case for Norway spruce and white/black spruce, although we have not directly worked with those species in the past.

  1. 411-418, giving only three of these scaffolds showed signatures of sweeps and polygenic adaptation in the same regions or very close genomic location within the scaffolds, the suggestion that “the presence of on-going directional or diversifying natural selection acting on genes involved in key metabolic processes, stress, transport and reproduction in both coast redwood and giant sequoia ….” may be too stretched.

>> This paragraph refers to only the selective sweeps analyses. As stated in lines 255-263 of the original manuscript, genomic regions containing sweep signatures were found throughout the genomes of both giant sequoia and coast redwood supporting our conclusion in lines 411-418.

Round 2

Reviewer 2 Report

I am happy with the revision and it should be accepted for publication

Author Response

Thanks